# Unlocking Slot Attention by Changing Optimal Transport Costs

**Yan Zhang**[*1]  **David W. Zhang**[*2]

**Simon Lacoste-Julien**[1,3,4]  **Gertjan J. Burghouts**[5]  **Cees G. M. Snoek**[2]

Samsung - SAIT AI Lab, Montreal[1]  University of Amsterdam[2]
Mila, Université de Montreal[3]  Canada CIFAR AI Chair[4]  TNO[5]

## Abstract

Slot attention is a successful method for object-centric modeling with images and videos for tasks like unsupervised object discovery. However, set-equivariance limits its ability to perform tiebreaking, which makes distinguishing similar structures difficult – a task crucial for vision problems. To fix this, we cast cross-attention in slot attention as an optimal transport (OT) problem that has solutions with the desired tiebreaking properties. We then propose an entropy minimization module that combines the tiebreaking properties of unregularized OT with the speed of regularized OT. We evaluate our method on CLEVR object detection and observe significant improvements from 53% to 91% on a strict average precision metric.

## 1 Introduction

Suppose we have an image containing two cats and one dog. Given a query like [cat, cat, dog], our task is to provide instance-specific information for each query element, such as their locations. When constructing a neural network to solve this problem, cross-attention is a natural choice to relate the queries to our image [20, 21]. With such a model, the dog can be located perfectly, but our two queries for the cats both end up with an undesirable result: the *average* of the two cats' positions. The problem is that with our model, multiple copies of the same query element *must* have the same result [23]. Models that rely on cross-attention, such as slot attention [15], can run into this issue when trying to extract objects from images and other data modalities.

This issue is present because cross-attention is set-equivariant (usually a desirable property), which Zhang et al. [23] show to be too restrictive for multisets (sets with repeated elements allowed which are prevalent in deep learning, background in Section 2). This manifests itself in two related problems:

1. **Soft assignments.** The model tends to mix several inputs into each slot rather than making a hard decision of one slot corresponding to exactly one input. This leads to difficulties when the information from each input must be kept distinct.

2. **Lack of tiebreaking.** Similar slots are processed similarly, so they will likely contain similar information. Multiple similar slots prefer to capture an average of the relevant inputs rather than each slot capturing a different input, which leads to the cat problem described above.

To avoid these issues, a property called exclusive multiset-equivariance is necessary [23]. So far, only models from the Deep Set Prediction Networks family [22, 23] are known to have this property, but they lack an object-centric inductive bias. Fortunately, introducing even a single exclusively multiset-equivariant module in a model is enough. We therefore develop a module that enhances cross-attention in order to obtain the desired equivariance property in the object-centric slot attention.

---

[*]Equal contribution

36th Conference on Neural Information Processing Systems (NeurIPS 2022).

**Contributions.** **1.** We make the connection that slot attention (SA) already uses an approximate regularized optimal transport (OT) (Section 3). The OT perspective is useful because some OT algorithms are able to break ties, which makes them useful for multiset-equivariance. This motivates variants where either the approximation or both approximation and regularization are removed, with the latter being exclusively multiset-equivariant. To avoid the speed penalty of unregularized OT, we introduce an entropy minimization module that modifies the cost for regularized OT while maintaining the desired equivariance. Our module is compatible with automatic differentiation and much faster than using an exact OT solver. **2.** We evaluate our model on the CLEVR object detection dataset [12] and compare it to default slot attention and several OT variants that we introduce as part of our motivation (Section 5). We obtain state-of-the-art results for the $AP_{\infty}$ metric and improve slot attention results from 53.1% to 91.1% on the $AP_{0.25}$ metric, all without large compromises in speed.

## 2 Background

Multisets are generalizations of sets by allowing repetitions of elements. In deep learning, both are represented as $\mathbb{R}^{n \times c}$ matrices with $n$ being the multiset size and $c$ the feature dimension per element. The uniqueness property of sets is rarely enforced in deep learning, so most models operate on multisets rather than sets. These models must then be careful to not rely on the arbitrary order of the $n$ elements. To *guarantee* this, they should satisfy certain equivariances.

### 2.1 Permutation equivariances

The standard definition of permutation-equivariant (**set-equivariant**) functions $f$ says that a permutation of the input should result in the same permutation of the output. With $\Pi$ as the space of $n \times n$ permutation matrices,

$$\forall \boldsymbol{X} \in \mathbb{R}^{n \times c}, \forall \boldsymbol{P} \in \Pi : \quad f(\boldsymbol{P}\boldsymbol{X}) = \boldsymbol{P}f(\boldsymbol{X}) \tag{1}$$

However, this means that a set-equivariant function must always produce the same result when there are equal inputs [23]: $f([\boldsymbol{a}, \boldsymbol{a}]) = [\boldsymbol{c}, \boldsymbol{d}]$ is not possible for $\boldsymbol{c} \neq \boldsymbol{d}$. Zhang et al. [23] therefore introduce a more appropriate equivariance for multisets, **multiset-equivariance**:

$$\forall \boldsymbol{X} \in \mathbb{R}^{n \times c}, \forall \boldsymbol{P}_1 \in \Pi, \exists \boldsymbol{P}_2 \in \Pi : \quad f(\boldsymbol{P}_1\boldsymbol{X}) = \boldsymbol{P}_2 f(\boldsymbol{X}) \tag{2}$$

This relaxation of set-equivariance makes $f([\boldsymbol{a}, \boldsymbol{a}]) = [\boldsymbol{c}, \boldsymbol{d}]$ possible. In practice, the primary benefit is that when two elements in the input set are *similar*, they do not have to result in similar outputs. All set-equivariant models are also multiset-equivariant, which means that only models that are *exclusively multiset-equivariant* (multiset-equivariant, but not set-equivariant) are capable of modeling $f([\boldsymbol{a}, \boldsymbol{a}]) = [\boldsymbol{c}, \boldsymbol{d}]$ for differing $\boldsymbol{c}$ and $\boldsymbol{d}$ successfully. Unfortunately, most operations in the multiset learning literature are set-equivariant and thus unable to break ties.

### 2.2 Slot attention

Slot attention (SA) [15] can be used to allocate objects in an image to a multiset of "slots". The module alternates applying cross-attention between the multiset of input features and the multiset of slot features to compute updates for each slot, and using a GRU [6] to apply the respective update to each slot. For example, the multiset of feature vectors in an image (corresponding to grid positions in the feature map) can be effectively summarized into a much smaller multiset of slots – each containing information about an object in the image.

Slot attention takes inputs $\boldsymbol{X} \in \mathbb{R}^{n \times c}$, then randomly initializes slots $\boldsymbol{Z}^{(0)} \in \mathbb{R}^{m \times d}$ and runs the following iteration of cross-attention with key, query, and value matrices followed by a GRU update.

$$\boldsymbol{Q}^{(n)} = \boldsymbol{Z}^{(n)}\boldsymbol{W}_Q, \quad \boldsymbol{K} = \boldsymbol{X}\boldsymbol{W}_K, \quad \boldsymbol{V} = \boldsymbol{X}\boldsymbol{W}_V \tag{3}$$

$$\boldsymbol{A}^{(n)} = \texttt{normalize}(\mathrm{softmax}(\boldsymbol{Q}^{(n)}\boldsymbol{K}^{\top})) \tag{4}$$

$$\boldsymbol{Z}^{(n+1)} = \mathrm{GRU}(\boldsymbol{Z}^{(n)}, {\boldsymbol{A}^{(n)}}^{\top}\boldsymbol{V}) \tag{5}$$

In Locatello et al. [15], $\mathrm{softmax}$ sets each sum over the $m$ slots to be 1 while $\texttt{normalize}$ sets each sum over the $n$ inputs to be 1. All operations used in slot attention are set-equivariant with respect to the slots, which makes the model set-equivariant [15]. In this paper, we introduce a module to make slot attention exclusively multiset-equivariant, which allows it to break ties between similar slots.

# 3 Connecting slot attention to optimal transport

**Entropy-regularized optimal transport.** A critical component when applying cross-attention in slot attention is the normalization of the attention matrix. Equation 4 first exponentiates all the entries, then normalizes one dimension of the attention matrix to sum to 1, then the other dimension to sum to 1. This sequence of operations is also known as applying the Sinkhorn algorithm for a single step. The Sinkhorn algorithm solves the entropy-regularized optimal transport problem [7] by repeatedly alternating these two normalizations. This results in a doubly-stochastic matrix at convergence. We can therefore think of Equation 4 as approximating this entropy-regularized optimal transport (by using only one Sinkhorn iteration) to determine how the information from the input should be associated with the slots. This connection between attention and optimal transport has also been made by Sander et al. [19]. A simple extension is thus to consider slot attention (SA) using more than one iteration of the Sinkhorn algorithm, which we will refer to as **SA-Sinkhorn**. In the context of this algorithm, computing the dot products in Equation 4 is equivalent to computing the cost matrix $C$ using the Euclidean distance [19]:

$$C_{ij} = ||\boldsymbol{Q}_i - \boldsymbol{K}_j||^2 \tag{6}$$

$$\boldsymbol{A} = \text{sinkhorn}(\boldsymbol{C}) \tag{7}$$

This variant brings us closer to optimal transport, though it is still restricted by set-equivariance. One additional detail is that the Sinkhorn algorithm must handle non-square matrices since the number of slots is usually much smaller than the number of inputs; naïvely applying the algorithm on such matrices does not converge. We describe the details of how to handle this in Appendix A as they are not important to the following discussion.

**Optimal transport.** Given the step toward optimal transport taken through the full use of the Sinkhorn algorithm, a straightforward question is whether other OT algorithms make sense in the context of slot attention. A benefit of OT without regularization is that they can be *exclusively multiset-equivariant*. By replacing entropy-regularized OT with unregularized OT, we can make slot attention exclusively multiset-equivariant to avoid the issues we pointed out in Section 1. Therefore, we propose the **SA-EMD** (Earth Mover's Distance) variant, wherein we use the EMD algorithm by Bonneel et al. [4] that is part of the POT package [8]. This does not come without its own problems however. By nature of solving an unregularized OT problem, its gradients are piecewise constant, which makes learning difficult. There are several methods for estimating the gradient of such an $\arg\min$ operation [10, 9, 1, 17]. In practice, we observe that the gradient of the Sinkhorn OT is also a good descent direction for EMD. We find that we obtain the best empirical results through:

$$\boldsymbol{A} = \text{sinkhorn}(\boldsymbol{C}) + \text{emd}(\boldsymbol{C}) \tag{8}$$

Another problem is that this tends to be rather slow. Standard solvers use a network simplex algorithm (a variant of the simplex algorithm for graphs), which is difficult to efficiently parallelize on GPUs.

Can we get the benefits of unregularized OT with its exclusive multiset-equivariance, while still being fast and differentiable like entropy-regularized OT? We answer this question in the following.

## 3.1 Minimizating the entropy of Sinkhorn

Previously, we focused on different ways of turning costs $C$ into a transport map $A$. Now, we focus on *changing the costs themselves*, then simply using the Sinkhorn algorithm on these new costs. A key difference between unregularized OT and entropy-regularized OT is of course the entropy in the resulting transport map. *The idea is to change the costs in such a way so that even after entropy-regularized OT, the entropy remains low.* The low entropy allows the tiebreaking necessary for exclusive multiset-equivariance. However, this adjustment to the costs should end up in a transport map that is still related to the original transport map before adjustment. This gives us the following optimization problem to minimize the entropy $H$:

$$\text{ME}(\boldsymbol{C}) = \underset{\boldsymbol{C}'}{\arg\min} \, H(\text{sinkhorn}(\boldsymbol{C}')) + \lambda||\text{sinkhorn}(\boldsymbol{C}')\boldsymbol{S} - \text{sinkhorn}(\boldsymbol{C})||^2 \tag{9}$$

$$\boldsymbol{A} = \text{sinkhorn}(\text{ME}(\boldsymbol{C})) \tag{10}$$

We call this variant **SA-ME** (Minimize Entropy). The first term aims to change the cost so that the resulting transport map after the Sinkhorn algorithm has low entropy (preferring 0s and 1s). The second term (not always required in practice[2]) makes sure that this redistribution of costs is allowed

---

[2]Empirically, $\lambda$ can be 0 when $\boldsymbol{C}'$ is initialized close to $\boldsymbol{C}$, making it purely a minimization of entropy.

for similar slots, but not allowed for dissimilar slots (we elaborate on the details of $\boldsymbol{S}$ in Appendix B). The final transport map is calculated from this new cost efficiently using the Sinkhorn algorithm. The result is that among similar slots, the costs are changed to make the transport map look more like the output of unregularized OT.

To solve this optimization problem, we propose to use gradient descent for a small fixed number of steps. Differentiating through this can be done with standard automatic differentiation. While the process described so far does not feature any tiebreaking explicitly and would struggle to minimize entropy successfully, we can simply add a small amount of noise at the start of the optimization: $\boldsymbol{C}'^{(0)} = \boldsymbol{C} + \boldsymbol{\epsilon}, \boldsymbol{\epsilon}_{ij} \sim \mathcal{N}(0, 0.001)$. An important detail is then to normalize the gradient to have a fixed norm: the small amount of noise is amplified when slots are similar in order to break ties, but without having to resort to large learning rates and impacting the stability of optimization. We repeat the following for $N$ steps:

$$\boldsymbol{C}'^{(n+1)} = \boldsymbol{C}'^{(n)} - \lambda \frac{\nabla_{\boldsymbol{C}'^{(n)}} h\left(\boldsymbol{C}'^{(n)}\right)}{\left\|\nabla_{\boldsymbol{C}'^{(n)}} h\left(\boldsymbol{C}'^{(n)}\right)\right\|}, \quad h(\boldsymbol{C}'^{(n)}) = H(\mathrm{sinkhorn}(\boldsymbol{C}'^{(n)})) \qquad (11)$$

**Equivariance.** SA-ME is now exclusively multiset-equivariant: it is multiset-equivariant because all the individual operations are multiset-equivariant, but it is not set-equivariant since similar slots are no longer guaranteed to receive similar transport maps due to the noise and the subsequent optimization breaking ties. This gives our method more representational power on multisets than standard slot attention and SA-Sinkhorn, as it is no longer restricted by set-equivariance.

A useful side-effect is that random initialization of slots is no longer necessary. Since ties can be broken by this entropy minimization, it is no problem to initialize all slots to be the same vector. In standard slot attention, the amount of noise in $\boldsymbol{Z}^{(0)}$ needs to be just right: too low, and the set-equivariant model has difficulties breaking ties between these similar slots; too high, and the model can become unreliable from the noisiness. In contrast, a tiny amount of noise (as long as it is above machine precision after $\mathrm{sinkhorn}$) in $\boldsymbol{C}'$ is sufficient for tiebreaking – preventing collapsing of slots – and any other effect of the noise can be optimized away through the gradient descent.

**Computation.** In comparison to SA-EMD, SA-ME is significantly faster. It only relies on evaluating the Sinkhorn algorithm for a relatively small number of optimization steps (we find little benefit above four steps), so the loss of speed compared to SA-Sinkhorn is limited while giving slot attention the benefits of exclusive multiset-equivariance of SA-EMD. Another benefit over SA-EMD is that gradient computation is simple, since we can fully rely on automatic differentiation instead of having to estimate gradients for the black-box EMD solver in SA-EMD.

## 4 Related work

An important consideration is how our entropy minimization method is different from simply reducing the temperature of the Sinkhorn algorithm, which also results in lower entropy transport maps. The issue with lowering temperature is that it runs into similar problems as using exact OT solvers in terms of pathological gradients. A trade-off needs to be found between the temperature being too high (too high entropy, cannot separate similar rows/columns) and the temperature being too low (gradients become piecewise constant, learning is difficult). In contrast, SA-ME does not feature such a trade-off. By using gradient descent with normalized gradients within the module, even extremely similar rows and columns can be separated easily. We use one differentiable procedure (our module) to adjust the costs, then another differentiable procedure (Sinkhorn) to obtain the low entropy map, all without needing to reduce the Sinkhorn temperature.

Another approximation of optimal transport can be obtained through the Sliced Wasserstein Distance (SWD) [3]. SWD performs tiebreaking through the use of numerical sorting and is thus exclusively multiset-equivariant, but it lacks precise 1-to-1 associations between inputs and slots. This can especially be a problem with varying input sizes. We tried approaches based on SWD as a replacement for standard cross-attention but did not obtain any competitive results.

In a similar direction to Sinkhorn which performs entropy-regularized OT, Blondel et al. [2] study L2-regularized OT problems. While their solver is faster than unregularized OT and obtains lower entropy solutions than Sinkhorn, similarly to Sinkhorn the convexity of the problem means that it cannot break ties effectively on its own, even with noise. In SA-ME, we can replace Sinkhorn with this method, but we found that it was too slow comparatively.

Table 1: Results on CLEVR object property multiset prediction, average precision (AP) in % (mean ± standard deviation) over 5 random seeds, higher is better. All SA results are based on our re-implementation. SA (original) results copied from Locatello et al. [15], iDSPN results from Zhang et al. [23]. Models with † use the improvement proposed by Chang et al. [5].

| Model | $AP_\infty$ | $AP_1$ | $AP_{0.5}$ | $AP_{0.25}$ | $AP_{0.125}$ | $AP_{0.0625}$ | Time |
|---|---|---|---|---|---|---|---|
| iDSPN [23] | $98.8_{\pm 0.5}$ | $98.5_{\pm 0.6}$ | $98.2_{\pm 0.6}$ | $\mathbf{95.8_{\pm 0.7}}$ | $\mathbf{76.9_{\pm 2.5}}$ | $\mathbf{32.3_{\pm 3.9}}$ | — |
| SA (original) [15] | $94.3_{\pm 1.1}$ | $86.7_{\pm 1.4}$ | $56.0_{\pm 3.6}$ | $10.8_{\pm 1.7}$ | $0.9_{\pm 0.2}$ | — | — |
| SA | $89.1_{\pm 1.2}$ | $85.7_{\pm 1.0}$ | $73.3_{\pm 1.2}$ | $35.4_{\pm 1.5}$ | $9.0_{\pm 0.8}$ | $2.0_{\pm 0.3}$ | 2.4 h |
| SA-Sinkhorn | $95.6_{\pm 1.0}$ | $94.0_{\pm 1.1}$ | $84.5_{\pm 1.7}$ | $41.3_{\pm 3.0}$ | $10.4_{\pm 0.7}$ | $2.5_{\pm 0.4}$ | 2.5 h |
| SA-EMD | $99.2_{\pm 0.2}$ | $98.7_{\pm 0.4}$ | $97.0_{\pm 0.8}$ | $82.4_{\pm 1.2}$ | $34.0_{\pm 2.2}$ | $8.3_{\pm 0.9}$ | 9.7 h |
| **SA-ME** | $99.2_{\pm 0.3}$ | $99.1_{\pm 0.3}$ | $98.8_{\pm 0.5}$ | $88.3_{\pm 0.8}$ | $40.8_{\pm 1.0}$ | $10.6_{\pm 0.3}$ | 2.5 h |
| SA† | $94.3_{\pm 0.4}$ | $85.7_{\pm 1.6}$ | $77.2_{\pm 1.5}$ | $53.1_{\pm 2.7}$ | $16.7_{\pm 1.8}$ | $4.0_{\pm 0.7}$ | 2.2 h |
| SA-Sinkhorn† | $98.9_{\pm 0.2}$ | $97.7_{\pm 0.5}$ | $95.2_{\pm 0.9}$ | $83.3_{\pm 0.8}$ | $38.5_{\pm 2.0}$ | $10.0_{\pm 1.4}$ | 2.3 h |
| SA-EMD† | $99.3_{\pm 0.3}$ | $98.1_{\pm 0.4}$ | $95.9_{\pm 0.8}$ | $85.8_{\pm 1.1}$ | $42.0_{\pm 2.0}$ | $11.4_{\pm 1.3}$ | 9.3 h |
| **SA-ME†** | $\mathbf{99.4_{\pm 0.1}}$ | $\mathbf{99.2_{\pm 0.2}}$ | $\mathbf{98.9_{\pm 0.2}}$ | $91.1_{\pm 1.1}$ | $47.6_{\pm 0.8}$ | $12.5_{\pm 0.4}$ | 2.4 h |

## 5 CLEVR Experiments

CLEVR [12] is a synthetic dataset containing images with up to ten objects in a 3d scene. Each object is sampled with varying sizes, materials, shapes, and colors. The task is to predict the multiset of objects with their properties and 3d position. Following Zhang et al. [22], we evaluate using average precision (AP) at different distance thresholds for the 3d coordinates of the predicted objects.

**Results.** Table 1 shows the various proposed ways of incorporating OT into slot attention, and also results for original slot attention and iDSPN for reference. In summary, our SA-ME achieves the best slot attention results without increasing run time by much. These results are followed by SA-EMD, which has slightly worse results (possibly due to inherently imprecise gradient estimation) but takes over three times longer to train. Interestingly, SA-ME and SA-EMD benefit relatively little from applying Chang et al. [5] (models with †) compared to SA and SA-Sinkhorn. SA-Sinkhorn† performs respectably compared to SA† considering its similarities in complexity and run time.

There are several potential reasons why our slot attention variant is still far from iDSPN results for the strictest thresholds like $AP_{0.0625}$. Our training scheme matches iDSPN in the number of epochs, but our model has not yet fully converged, so different hyperparameters such as more epochs should be beneficial. The image backbone is also smaller, and there may be a different inductive bias that prioritizes $AP_\infty$ over $AP_{0.0625}$ for SA. In general, slot attention through the use of attention has the benefit of not needing to compress the input into a single vector like iDSPN. The resulting object-centric inductive bias and the relative simplicity have allowed for wider adoption and success of slot attention over iDSPN [13, 11, 14, 18], which makes our improvements to SA meaningful.

## 6 Discussion

We introduced several variants of slot attention that allow it to break ties in its slots which allows for better modeling of objects. In particular, entropy minimization looks to be a promising approach to give cross-attention and thus slot attention the ability to be exclusively multiset-equivariant. While we already see excellent results on CLEVR, the general applicability of our model is uncertain without a more varied set of experiments. For example, we aim to evaluate on more tasks that slot attention has been shown to work well at, such as unsupervised and weakly-supervised object discovery.

We tried several alternatives with different types of objectives other than entropy such as $-(x-0.5)^2$. Many of these variants do not improve results, but also do not make them worse. This suggests that this idea of minimizing the *entropy* of entropy-regularized OT may not be crucial in itself, even if it seems elegant. Distilling what is absolutely necessary about our method will lead to simpler models and improved understanding. We also aim to better understand the interplay between Chang et al. [5] († models) and optimal transport, since they do not seem to be orthogonal improvements on CLEVR.

While our module is in principle applicable to any use case of self- or cross-attention, we have so far not seen any benefits in a few preliminary experiments with vision transformers. We believe that this is due to the lesser importance of keeping object identities separate, particularly in tasks that are not obviously object-centric. Understanding in which cases our method is beneficial a priori requires further investigation.

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

## A    Marginals in Sinkhorn and EMD

As we mention in the main text, we need to account for the (common) case of the number of inputs $n$ and the number of slots $m$ differing, i.e. with a cost matrix $\boldsymbol{C} \in \mathbb{R}^{n \times m}$. The problem is that it is impossible to make every row and every column of the transport map sum to 1 when the number of rows and columns is different. Fortunately, there is standard practice for how to deal with this case in optimal transport [16, 7, 4]. We can define non-negative marginals $\boldsymbol{a} \in \mathbb{R}^n$ and $\boldsymbol{b} \in \mathbb{R}^m$ that specify the row and column sums of the transport map respectively. If $\sum_i \boldsymbol{a}_i = \sum_j \boldsymbol{b}_j$, then convergence is as normal.

In our case, we learn $\boldsymbol{a} = m \cdot \text{softmax}(h(\boldsymbol{Z}))$ with a neural network $h : \mathbb{R}^c \to \mathbb{R}$ that is shared across the $n$ input elements and simply set $\boldsymbol{b} = \boldsymbol{1}$. Our $\boldsymbol{a}$ allows the model to put focus on important input elements (e.g. the inputs corresponding to objects) and ignore unimportant input elements (e.g. the inputs corresponding to the background). Our $\boldsymbol{b}$ specifies that every slot should be equally important. Since the softmax sums to one, we have $\sum_i \boldsymbol{a}_i = \sum_j \boldsymbol{b}_j = m$ so there is no problem with convergence.

For the Sinkhorn algorithm, it now repeatedly alternates normalizing all the rows to sum to $\boldsymbol{a}$, then all the columns to sum to $\boldsymbol{b}$. For the EMD solver that we use [4], these marginals are standard parameters in the algorithm. In the main text, we omit these marginals whenever we refer to sinkhorn or emd for simplicity of notation.

## B    Explanation of similarity matrix S

The following discussion is not critical to understanding the main text, since we find empirically that with the right initialization (e.g. $\boldsymbol{C}' = \boldsymbol{C} + \epsilon$ with $\epsilon \sim \mathcal{N}(0, 10^{-3}\boldsymbol{I})$), $\lambda$ can be safely set to 0. We only find that $\lambda > 0$ is necessary with an initialization such as with $\epsilon \sim \mathcal{N}(0, \boldsymbol{I})$. As we mention in the main text, the amount of noise is not important as long as it remains above machine precision after applying sinkhorn, which can be easily checked a priori. This appendix serves to explain the purpose of the case when $\lambda > 0$, even if we find in experiments so far that it is not necessary in practice.

The idea of $\boldsymbol{S}$ is to allow costs to be freely changed among similar slots, but disallow this for dissimilar slots. The aim of $||\text{sinkhorn}(\boldsymbol{C}')\boldsymbol{S} - \text{sinkhorn}(\boldsymbol{C})||^2$ is to make sure that $\text{sinkhorn}(\boldsymbol{C}')$ looks the same as $\text{sinkhorn}(\boldsymbol{C})$ after allowing weight in the transport map to be moved around among similar slots.

**Example**    Consider the case where we have three slots: $\boldsymbol{Z} = [\boldsymbol{x}, \boldsymbol{x}, \boldsymbol{y}]$ and three inputs $\boldsymbol{X} = [\boldsymbol{\alpha}, \boldsymbol{\beta}, \boldsymbol{\gamma}]$. Let us assume for this example that the cost matrix prefers associating $\boldsymbol{\gamma}$ with $\boldsymbol{y}$ and both $\boldsymbol{\alpha}$

and $\boldsymbol{\beta}$ with $\boldsymbol{x}$. Computing unregularized OT solutions would therefore give us either of two solutions:

$$\boldsymbol{T}_1 = \begin{bmatrix} 1 & 0 & 0 \\ 0 & 1 & 0 \\ 0 & 0 & 1 \end{bmatrix} \quad \text{or} \quad \boldsymbol{T}_2 = \begin{bmatrix} 0 & 1 & 0 \\ 1 & 0 & 0 \\ 0 & 0 & 1 \end{bmatrix} \tag{12}$$

However, the Sinkhorn algorithm is unable to break the tie between the two $\boldsymbol{x}$ slots, so even with a temperature approaching 0, we obtain the following result:

$$\text{sinkhorn}(\boldsymbol{C}) = \begin{bmatrix} 0.5 & 0.5 & 0 \\ 0.5 & 0.5 & 0 \\ 0 & 0 & 1 \end{bmatrix} \tag{13}$$

Suppose we have a similarity matrix $\tilde{\boldsymbol{S}} \in \mathbb{R}^{m \times m}$ ($m$ is the number of slots) that measures pairwise similarities ranging from 0 to 1:

$$\tilde{\boldsymbol{S}} = \begin{bmatrix} 1 & 1 & 0 \\ 1 & 1 & 0 \\ 0 & 0 & 1 \end{bmatrix} \tag{14}$$

The first two slots are similar amongst themselves but dissimilar to the $\boldsymbol{y}$ slot. If we normalize each column of $\tilde{\boldsymbol{S}}$ to sum to 1 to obtain $\boldsymbol{S}$, then we see the following:

$$\underbrace{\begin{bmatrix} 1 & 0 & 0 \\ 0 & 1 & 0 \\ 0 & 0 & 1 \end{bmatrix}}_{\boldsymbol{T}_1} \underbrace{\begin{bmatrix} 0.5 & 0.5 & 0 \\ 0.5 & 0.5 & 0 \\ 0 & 0 & 1 \end{bmatrix}}_{\boldsymbol{S}} = \underbrace{\begin{bmatrix} 0.5 & 0.5 & 0 \\ 0.5 & 0.5 & 0 \\ 0 & 0 & 1 \end{bmatrix}}_{\text{sinkhorn}(\boldsymbol{C})} \tag{15}$$

$$\underbrace{\begin{bmatrix} 0 & 1 & 0 \\ 1 & 0 & 0 \\ 0 & 0 & 1 \end{bmatrix}}_{\boldsymbol{T}_2} \underbrace{\begin{bmatrix} 0.5 & 0.5 & 0 \\ 0.5 & 0.5 & 0 \\ 0 & 0 & 1 \end{bmatrix}}_{\boldsymbol{S}} = \underbrace{\begin{bmatrix} 0.5 & 0.5 & 0 \\ 0.5 & 0.5 & 0 \\ 0 & 0 & 1 \end{bmatrix}}_{\text{sinkhorn}(\boldsymbol{C})} \tag{16}$$

This means that $\text{sinkhorn}(\boldsymbol{C}') = \boldsymbol{T}_1$ and $\text{sinkhorn}(\boldsymbol{C}') = \boldsymbol{T}_2$ are both valid solutions for the minimization of $||\text{sinkhorn}(\boldsymbol{C}')\boldsymbol{S} - \text{sinkhorn}(\boldsymbol{C})||^2$. Note that any other permutation matrix for $\boldsymbol{T}$ (i.e. one where there isn't a 1 in the bottom right corner) would not be a valid solution. This restricts Equation 9 to only consider transport maps that are convex combinations of $\boldsymbol{T}_1$ and $\boldsymbol{T}_2$ for this example, with the entropy minimization preferring $\boldsymbol{T}_1$ and $\boldsymbol{T}_2$ specifically. The small amount of noise in the $\boldsymbol{C}'$ initialization arbitrarily makes it prefer one of the two.

**Definition** We define the similarity matrix $\tilde{S}_{ij} = g(\boldsymbol{Z}_i, \boldsymbol{Z}_j)$, where $g : \mathbb{R}^c \times \mathbb{R}^c \to \mathbb{R}$ is a small neural network that takes pairs of slots as input and produces a similarity score as output. We then normalize each column of $\tilde{\boldsymbol{S}}$ to sum to 1 by applying softmax on each column.

$$\boldsymbol{S} = \text{softmax}(\tilde{\boldsymbol{S}}) \tag{17}$$

If we set up a training task for the example described above, we observe that $\boldsymbol{S}$ is learned to be virtually the same as the $\boldsymbol{S}$ we use in the example.

