# OpenReview forum: "Unlocking Slot Attention by Changing Optimal Transport Costs"
_NeurIPS.cc/2022/Workshop/nCSI — nCSI WS @ NeurIPS 2022 Oral_

### Official Review · Reviewer_5gjB · 2022-10-14
**A well-written paper relating cross-attention and the optimal transport problem to solve an interesting problem with slot attention**

**Rating:** 2
**Confidence:** 2

**Review:**

**Summary of the Paper:**
This paper looks at a problem with slot attention (SA) where similar objects are not treated independently and instead combined to one result which does not represent either one of the objects well. This is argued to be a consequence of the *set-equivariance* property. Therefore, the authors aim to make slot attention *exclusively multiset-equivariant*, i. e. allowing similar inputs to have non-similar results. To this end, they relate the calculation of the attention matrix $A$ in SA to the problem of (entropy-regularized) optimal transport (OT). They introduce several approaches using the Sinkhorn algorithm to obtain $A$. In trying to combine the advantages of those, they end up with **SA-ME** which calculates $A$ through an entropy-regularized OT problem and also minimizes the entropy to allow for "tiebreaking" (and therefore multiset-equivariance). Experiments show SA-ME to have better results than other slot attention approaches without a significant increase in computation time.

**Strengths:**
I like the motivation behind the paper and how it is structured to make the thought process, advantages, and disadvantages behind the introduced SA approaches clear. It is very well-written. Overall, the goal of creating multiset-equivariant methods appears to be a worthwhile endeavor. I also like how the authors discuss the results and include iDSPN despite its better performance on the more strict thresholds. Here, I also agree with the authors that the proposed methodology is still meaningful, both because it increases results compared to SA in general and compared to iDSPN for some thresholds in particular, and because I think the motivation and idea behind this methodology might also be useful elsewhere. Although not very important, I also really like the explanation and example for the similarity matrix $S$ in the appendix.

**Weaknesses/Possible Improvements:**
The paper might benefit from (although not require) one or two figures illustrating motivation and results. I would have liked to know more insights on why iDSPN performed better on the stricter thresholds and if SA-ME could get as good results if it was closer to convergence. It would have been nice if there was some experiment which goes back to the motivation, for example a query of two identical or very similar objects in the CLEVR dataset and comparing this to standard SA. Even just a small qualitative experiment here could help to highlight the benefit of the proposed method in such a scenario (again, a figure would have been nice). However, I do acknowledge the page limit and as a result can understand the decisions made, shortening other parts might be difficult.

Is there a specific reason why the re-implementation of SA is so much better than the original SA results (stricter thresholds)? I also find the question in the last sentence (line 211) quite interesting but it is fair to leave this for future work.

Line 138/139: I believe it should say "tie breaking optimizations" instead of "optimization breaking ties".

**Review Summary:**
To sum up my review, I can say that while I was not very familiar with set-/multiset-equivariance, the OT problem and the Sinkhorn algorithm, the paper itself and the cited references made things clear to me. Although I am no expert on this topic, I now feel confident in my assessment and suggest accepting this paper to the workshop. While I would have liked to see a bit more regarding the results, I really like the motivation, methodology, and the explanation of the methodology and think that these points warrant accepting this paper.

---

### Official Review · Reviewer_NRVZ · 2022-10-14
**An intruiging theoretical connection leads to convincing empirical results**

**Rating:** 3
**Confidence:** 2

**Review:**

The paper investigates methods for making slot attention exclusively multiset-equivariant, a property which allows them to map identical queries to different outputs. Starting with the observation that the normalization steps of standard slot attention resemble an iteration of the Sinkhorn algorithm for the entropy-regularized optimal transport problem, various methods are proposed to approximate solving the unregularized optimal transport problem within slot attention, yielding the desired properties. The methods are evaluated on CLEVR multiset property prediction.

Pros:
 - The paper makes a very intriguing and non-obvious connection between the attention matrix of slot attention and optimal transport problems, that I think will be of interest to the community even by itself.
 - The proposed slot attention variants represent a convincing exploration of this connection: First, the normalization steps are repeated, resembling the Sinkhorn algorithm. Then, solving the unregularized OT problem, and a fast approximation thereof is proposed.
 - The empirical results support the authors' claim that multiset prediction is a weak point of slot attention which can be improved via exclusively multiset-equivariant components.
 - The paper is well written and easy to follow.

Cons (which the authors do a good job of identifying themselves in the discussion section):
 - While the connection to OT has led to promising results, some of the choices made along the way appear somewhat ad hoc (e.g. in Eq. 8 and 9), and call for further investigation.
 - Additional experiments other than CLEVR would further support the hypothesis of the paper.

Overall, I think this is a very enlightening paper that will be of great interest to the community.

---

### Meta-Review · Area_Chair_PXAm · 2022-10-18

**Recommendation:** 3
**Confidence:** 3

**Metareview:**

The paper makes a relevant contribution: an entropy-regularized minimization to allow for tiebreaking (multiset-equivariance) including experimental results showing better results than other slot attention approaches without a significant increase in computation time. The reviewers are enthusiastic about the paper. The paper should also elicit some useful discussion of pros and cons as well as future work.

---

### Decision · Program_Chairs · 2022-10-20

Accept (Oral)